

# Spatial Disparities of Ozone Pollution in the Sichuan Basin Spurred by an Extreme Heatwave

Nan WANG[1*], Yunsong DU[1], Dongyang CHEN[1], Haiyan MENG[1], Xi CHEN[2], Li ZHOU[1], Guangming SHI[1], Yu ZHAN[1], Miao FENG[3], Wei LI[3], Mulan CHEN[4], Zhenliang LI[4], Fumo YANG[1*]

[1]College of Carbon Neutrality Future Technology, Sichuan University, Chengdu, China
[2]Institute of Mass Spectrometry and Atmospheric Environment, Guangdong Provincial Engineering Research Center for On-line Source Apportionment System of Air Pollution, Jinan University, Guangzhou, PR China
[3]Chengdu Academy of Environmental Sciences, Chengdu, 610072, China
[4]Chongqing Research Academy of Eco-Environmental Sciences, Chongqing, 401147, China

Correspondence to fmyang@scu.edu.cn, nan.wang@scu.edu.cn

**KEYWORDS**: ozone pollution, heatwave, spatial disparity, Sichuan Basin, ozone formation mechanism



## Abstract

Under the influence of climate change, the increasing occurrence of extreme weather events, such as heatwaves, has led to an enhanced frequency of ozone ($O_3$) pollution issues. In August 2022, the Sichuan Basin (SCB), a typical large-scale geographical terrain located in southwestern China, experienced the most severe heatwave over the last 20 years. The heatwave led to substantial disparities in $O_3$ levels across the region. Here, by integrating observations, machine learnings and numerical simulations, we aim to understand the diverse $O_3$ formation mechanisms in two mega cities, Chengdu (western location) and Chongqing (eastern location). Observational data showed that Chengdu experienced a consecutive 17-day period of $O_3$ exceedance, in contrast to Chongqing, where $O_3$ concentrations remained below the standard. Meteorological and precursor factors were assessed, spotlighting high temperatures, intense solar radiation, and overnight accumulative pollutants as key contributors to $O_3$ concentrations. The interplay of isoprene, temperature, and $O_3$, alongside the observation-based box model and MEGAN simulations, underscored the significant role of intensified biogenic VOCs (BVOCs) on $O_3$ formations. Interestingly, Chongqing exhibited nearly double the BVOCs emissions of Chengdu, yet contributed less to $O_3$ concentrations. This discrepancy was addressed through CMAQ-DDM simulations and satellite diagnosis by investigating the $O_3$-$NO_x$-VOCs sensitivity. Notably, Chengdu displayed a VOCs-driven sensitivity, while Chongqing showed a transitional regime. Moreover, the regional transport also played a pivotal role in the spatial divergence of $O_3$ pollution. Cross-regional transport predominantly influenced Chongqing (contributing ~80%), whereas Chengdu was mainly affected by the emissions within the basin. The local accumulated pollutants gave rise to the atmospheric oxidizing capacity, resulting in a substantial photochemical contribution to $O_3$ levels (49.9 ppbv/hour) in Chengdu. This comparison of the difference provides the insights into the complex interplay of meteorology, natural emissions, and anthropogenic sources during heatwaves, guiding the necessity of targeted pollution control measures in regional scales.



## 1 Introduction

Ground-level ozone ($O_3$), formed through intricate photochemical reactions involving precursors like volatile organic compounds (VOCs) and nitrogen oxides ($NO_x$) under sunlight, is a prominent constituent of smog and a major contributor to poor air quality. Different from the protective role in the stratosphere, $O_3$ in the troposphere has garnered great attention due to its potential damage to human well-being and ecological systems (Krupa and Kickert, 1989; Schwela, 2000; Emberson et al., 2001; Xiao et al., 2021). The hazardous effects span across multiple domains, such as detrimental impact on human health, vegetation growth, and the climate. Addressing $O_3$ pollution is a complex endeavor, which mainly arises from the nonlinear relationship between $O_3$ and its precursors. Besides, the substantial influence of meteorological conditions adds another layer of intricacy to the challenge of managing $O_3$ pollution. Under global warming, the interplay of factors such as extreme weather events and elevated anthropogenic emissions have led to the frequent emergence of $O_3$ pollution, exacerbating air quality issues in urban areas worldwide.

Net $O_3$ production arises when the equilibrium between O3 and nitrogen oxides ($NO_x$), i.e., $NO + O_3 \rightarrow NO_2 + O_2$, is disrupted through the involvement of alkylperoxyl ($RO_2$) and hydroperoxyl ($HO_2$) radicals originating from oxidation of VOCs and carbon monoxide (CO). This intervention triggers the oxidation of NO to $NO_2$, ultimately resulting in the accumulation of $O_3$ through $NO_2$ photolysis (Jacob, 2000; Lelieveld and Dentener, 2000). Functioning as a pivotal role in photochemical reactions, VOCs have been identified as a crucial focal point for advancing efforts in the prevention and management of $O_3$ pollution (Jenkin and Clemitshaw, 2000). However, influenced by the diversity, abundance and reactivity of VOCs species, the spatial and temporal of VOCs characteristics depict regional disparities, adding difficulty in developing an effective strategy to reduce photochemical smog. Moreover, due to the dual roles of $NO_x$ in $O_3$ formation, where they enhance $O_3$ formation in low $NO_x$ environments and titrate $O_3$ in high $NO_x$ environments, reductions in VOCs must be examined along with the patterns of $NO_x$. Given the diverse energy structures in different regions, comprehending the regional





responsiveness of $O_3$-$NO_x$-VOCs sensitivity is essential. This is
particularly vital for elucidating non-linear relationship discrepancies
within regional contexts, which helps to advance the formulation of
effective emission reduction strategies.
$O_3$ pollution episodes are also closely related to meteorology. High
temperature, intensive solar radiation and light winds are found to be the
unfavorable weather conditions inducing photochemical pollutions (Ding
et al., 2017; Wang et al., 2017; Wang et al., 2022b). Generally, the impact
of meteorological conditions on $O_3$ is manifested through factors such as
changes in chemical reaction rates, dry/wet deposition, and atmospheric
transport. By objectively classifying pollution weather types, numerous
studies have summarized the typical weather conditions that lead to $O_3$
pollution. For example, high-pressure ridge, continental anticyclone and
the periphery of typhoons are the typical weather system conducing $O_3$
pollutions in east Asia (Mcelroy et al., 1986; Daum et al., 2003; Wang et
al., 2015). Besides, meteorology can also indirectly affect $O_3$ by
modulating natural emissions, such as BVOCs (biogenic VOCs) emissions
from vegetation and reactive nitrogen emissions from soil (Hall et al., 1996;
Saunier et al., 2017; Huang et al., 2018). For instance, a rise in temperature
can result in elevated emissions of BVOCs, thereby contributing to the
formation of $O_3$ (Wang et al., 2022b). With the influence of climate change,
there is an increasing frequency of extreme weather events, further
perturbating the natural emissions and finally exacerbating $O_3$ pollutions
(Lu et al., 2019).
The Sichuan Basin (SCB), encircled by the Qinghai-Tibet Plateau, Yungui
Plateau, and surrounding mountain ranges, stands as a notable hotspot for
atmospheric pollution within China. Two mega cities, Chengdu and
Chongqing, are situated in the SCB with populations exceeding 50 million.
In fact, a considerable amount of research on the pollution characteristics
of $O_3$ has been conducted in the SCB. For example, the characteristics of
$O_3$ and the precursors have been widely measured and analyzed (Zhao et
al., 2018; Qiao et al., 2019; Zhou et al., 2020; Chen et al., 2022).The
complicated coupling effect between the plateau-deep basin topography
and the unique meteorological conditions on atmospheric pollution have





been studied (Hu et al., 2022; Shu et al., 2022; Lei et al., 2023). The impact
of aerosol feedbacks on $O_3$ was also explored (Wang et al., 2020).
However, a limited focus has been placed on contrasting the varied
responses among different sites or cities within the basin. Exploring and
contrasting diverse mechanisms across multiple sites enriches our
comprehension and facilitates collaborative air pollution mitigation efforts
in a regional scale. In August 2022, the SCB experienced an exceptionally
rare heatwave, with monthly mean temperature ranking the highest over
the last two decades. As a result, the Chengdu Plain suffered from 17-day
consecutive $O_3$ pollution, whereas Chongqing remained good air quality.
Here, we combined field measurements, machine learning and numerical
simulations to elucidate the spatial disparities of $O_3$ pollution mechanism
within the SCB. This information has implication for better understanding
the meteorological contributions, discrepancy in $O_3$-$NO_x$-VOCs sensitivity,
and regional transport disparities between large urban areas, and provides
insights for regional joint control of $O_3$ pollution.

## 2 Method

### 2.1 Data Source

Data of atmospheric compositions, including $O_3$, $NO_x$ (NO and $NO_2$), CO,
$SO_2$, VOCs components and meteorological parameters were collected
from two in-situ observational sites. One was the Junping Street Station in
Chengdu and the other was the Academy of Environmental Sciences
Station in Chongqing. Both sites situated in the urban center of Chengdu
and Chongqing, representing the air quality of urban sites. Detailed
information of the measurements, such as monitoring instruments, data
coverage, and resolution were summarized in Table S1. Briefly, the
ambient concentrations of $O_3$, $NO_x$, CO and $SO_2$ were detected by
instruments produced by Thermo Scientific (Model 49i, 42i, 48i and 43i,
respectively). The species of VOCs were sampled by the GC955-611/811
Ozone Precursor Analyser produced by Synspec. Meteorological
parameters including temperature, relative humidity, wind speed and wind
direction at the same sites were concurrently measured by the mini-weather
stations (WS600-UMB in Chengdu and WS502-WTB100 in Chongqing).





All instruments were meticulously maintained and regularly calibrated.
Moreover, the air quality monitoring network established by the Ministry
of Ecology and Environment of China was employed to assess $O_3$ pollution
events in the SCB.

## 2.2 Stepwise Regression Analysis

We employed the stepwise regression analysis to assess the impact of
various meteorological factors on $O_3$ formation. This approach involves the
introduction of numerous input variables, with the method iteratively
selecting significant factors while eliminating non-significant ones,
ultimately resulting in the identification of a final set of critical factors.
Following this, we constructed a multivariate linear regression equation to
model $O_3$ concentration. In detail, meteorological parameters were
obtained from the fifth generation of the European Centre for Medium-
Range Weather Forecasts atmospheric reanalysis (ERA5). The selected
parameters included 10m u-component of wind (U10), 10m v-component
of wind (V10), vertical wind (w), boundary layer height (BLH), 2m
temperature (T2) and surface solar radiation (SSR). Given the high
correlation (R=0.85) between the diurnal variations of T2 and SSR during
the heatwave, it was challenging to distinguish the individual impacts of
T2 and SSR. As a pragmatic approach, we chose to combine them by
multiplying T2 with SSR, thereby examining the collective influence of
elevated temperatures and high solar radiation. Additionally, we also
incorporated previous night accumulative air pollutants, such as $O_3$
(ACCO3) and $NO_2$ (ACCNO2), as input parameters to investigate the
impact of pollutants being overnight accumulated on $O_3$ levels. The
machine learning-simulated $O_3$ concentrations were then validated against
observations, revealing a robust correlation (R > 0.91, P < 0.01) between
them (Fig S1). This result demonstrates the effectiveness of meteorological
and overnight accumulative factors in explaining a substantial portion of
$O_3$ concentrations.

## 2.3 Observation-based model (OBM)

In this study, an observation-based box model (OBM) configured with the
master chemical mechanisms (MCM v3.3.1) was employed to identify the



key VOCs species influencing $O_3$ (Jenkin et al., 2015; Bloss et al., 2005;
Saunders et al., 2003; Jenkin et al., 2003; Jenkin et al., 1997). The model
considered VOCs concentrations, trace gases ($O_3$, $NO_x$, CO, $SO_2$),
meteorological parameters, as well as the photolysis rates of $NO_2$ ($J_{NO_2}$)
from the in-situ sites in Chengdu and Chongqing. Observations were used
as constraints in the model and were averaged to represent the diurnal cycle
with a time resolution of 1 hour. The photolysis rates generated by the
model were adjusted based on the measured $J_{NO_2}$ values in order to
accurately simulate the photochemical reactions. The mean mixing ratios
of 46 VOCs species, including 20 alkanes, 11 alkenes, 1 alkyne (ethyne)
and 14 aromatics were listed in Table S2. The model started at 00:00 local
time (LT) and ran for a period of 24 hours. Prior to the formal calculation,
we conducted a spin-up run for 4 days with constraints representing the
diurnal cycle, allowing the unconstrained compounds (e.g., radicals and
HCHO) to reach steady states. Using the OBM simulation, the relative
incremental reactivity (RIR) method was applied to assess the sensitivity
of $O_3$ formation to individual precursor species (Cardelino and Chameides,
1995; Meng et al., 2023; Zhang et al., 2019; Xue et al., 2014; Zhu et al.,
2020). The calculation process can be expressed in Eq. (1).
$$RIR(X) = \frac{(P_{O_3}(X) - P_{O_3}(\Delta X))/P_{O_3}(X)}{\Delta C(X)/C(X)} \quad (1)$$

Here, $X$ represents a specific precursor of $O_3$. $P_{O_3}(X)$ and $P_{O_3}(\Delta X)$
represent the maximum simulated $O_3$ concentration based on measured
concentration and the concentration when the precursor levels change by
$\Delta X$. $\Delta C(X)/C(X)$ indicates the relative change of precursor $X$. In this study,
a reduction of 20% in precursor $X$ was selected to perform the RIR analysis.

### 209   2.4 Lagrangian Particulate Dispersion Modeling

We conducted backward Lagrangian particulate dispersion modeling
(LPDM) to ascertain the potential source regions for the air masses
observed at the monitoring stations. This approach involved employing the
hybrid single-particulate Lagrangian-integrated trajectory model
(HYSPLIT) driven by the ARL format Global Data Assimilation System
(GDAS) data. The LPDM was executed with a temporal resolution of 1



hour, releasing 3000 particulates at 100 meters above sea level from the
site and then tracking their backward movement for 72 hours. The
particulates' positions were calculated in both vertical and horizontal
dimensions, considering the impact of atmospheric advection and diffusion.
By analyzing the resulting data, we derived the "retroplume", which
indicates the spatial residence time of particulates and reflects the
distribution of surface probability or simulated air mass residence time.
This technique enabled us to diagnose whether the in-situ observation was
predominantly influenced by local emissions or regional transport.

## 2.5 Chemical transport modeling

A chemical transport model, WRF-MEGAN-CMAQ (Weather Research
Forecast – Model of Emissions of Gases and Aerosols from Nature –
Community Multiscale Air Quality), was employed to study the $O_3$
formation mechanism in the SCB. We adopted a two-nested domain, with
the outer domain covering most parts of east Asia (grid resolution of 36×
36 km) and the inner domain covering the southwestern China with the
SCB being focused (grid resolution of 12×12 km). The European Center
for Medium-Range Weather Forecasts (ECMWF) reanalysis data was used
as the initial and lateral boundary conditions of the WRF (version 3.9.1).
Carbon Bond Mechanism Version 6 and Aerosol Scheme 6 were used for
gas-phase and aerosol chemical simulations within the CMAQ model
(version 5.4), respectively. With regard to anthropogenic emissions, the
recently updated 2020-based MEIC emissions (Multi-resolution Emission
Inventory for China, developed by Tsinghua University) were used for
areas within China and the 2010-based MIX emissions (Li et al 2017) were
used for regions outside China. Both sets of the emissions have a horizontal
resolution of 0.25×0.25°, incorporating sectors such as transportation,
industry, power plant, residential and agriculture. Besides, natural
emissions were calculated using MEGAN model (version 2.1) driven by
the WRF simulated meteorology. The static input vegetation-related data
of MEGAN were updated by using the 2020-based the plant function type
(PFT) and leaf area index (LAI) retrieved from the MODIS (Moderate-
Resolution Imaging Spectroradiometer) products. More details of the





modeling configuration were summarized in Table S3.
In this study, we introduced the CMAQ-DDM (Decoupled Direct Method)
module to investigate the non-linear relationship between $O_3$ and its
precursors. Unlike the traditional brute force method (BFM) that involves
cutting or eliminating emissions from source regions (or sectors), which is
not only computationally intensive but also prone to uncertainties (due to
the intricate non-linear nature of $O_3$ chemistry), the DDM method offers a
more refined alternative. It enables accurate and computationally efficient
calculations of the sensitivity coefficients required for evaluating the
impact of parameter variations on output chemical concentrations
(Napelenok et al., 2008). Herein, both first-order and higher order
sensitivities were calculated to obtain the $O_3$–$NO_x$–VOCs sensitivities in
Chengdu and Chongqing. Furthermore, we also utilized the CMAQ-ISAM
(Integrated Source Apportion Method) technique, an innovative approach
for source tracing. This method enables us to trace and quantify the distinct
impacts on $O_3$ concentrations originating from specific source sectors,
emissions confined within designated geographical regions, as well as
effects arising from stratospheric and lateral boundary conditions (Kwok
et al., 2013). Through this approach, we calculated the separate influences
of anthropogenic and biogenic emissions on $O_3$ levels. We also assessed
the contributions of source regions to $O_3$ levels in Chengdu and Chongqing,
encompassing both local and regional influences. A map of source region's
classification in this study was provided in Fig S2.
We validated the performance of the WRF-MEGAN-CMAQ model using
surface network monitoring data. The time series and statistical outcomes
of the simulated and observed $O_3$ within the SCB are consolidated in Fig
S3. In general, the favorable alignment between observations and
simulations underscores the model's proficiency in accurately replicating
the magnitude and temporal variations of air pollutants.
## 3 Results and discussion
### 3.1 Regional disparity of $O_3$ between Chengdu and Chongqing

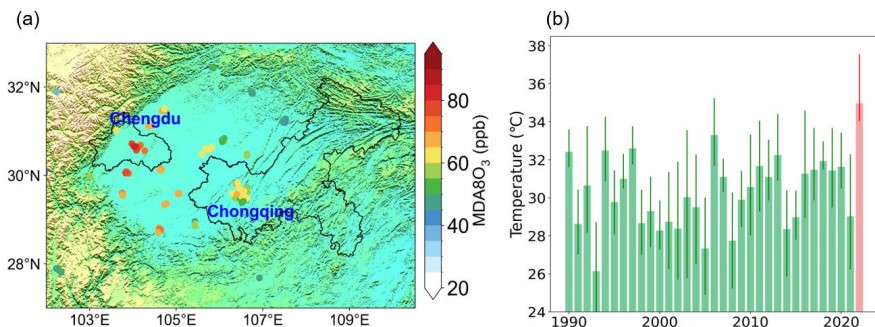

Fig 1 (a) Geographical distribution of Sichuan Basin with scattered averaged monthly MDA8 $O_3$ concentrations (data obtained from Ministry of Ecology and Environment of China). The black lines highlight the administrative border of Chengdu and Chongqing, respectively. (b) Historical monthly averaged daily-maximum air temperature (August) variation of the SCB from 1990 to 2022. The red bar highlights the extreme hot temperature in 2022.

August 2022 witnessed the SCB experiencing its hottest August in the last 20 years, with Chengdu and Chongqing reporting monthly mean temperatures soaring to 36.8°C and 40.3°C, respectively (Fig 1). Typically, the atmospheric conditions in the SCB are relatively stable due to the topography of the basin. This stability, in conjunction with elevated temperatures, tended to foster the occurrence of photochemical pollution (Zhao et al., 2018; Chen et al., 2022). However, during this historically unprecedented heatwave, $O_3$ levels exhibited substantial variations across the SCB. Observations revealed that $O_3$ concentrations surpassed China's Grade II standard (75 ppbv) in the western part of the SCB, notably in Chengdu. Conversely, significantly lower concentrations, well below the standard, were observed in the eastern region of the basin, particularly in Chongqing (Fig 1a). According to the network monitoring data, the average maximum daily 8-hour (MDA8h) $O_3$ concentration in Chengdu was measured at 75.1 ppbv. In contrast, the MDA8h $O_3$ concentration in Chongqing was recorded at 55.1 ppbv.

Based on the synoptic weather system, it could be found that the SCB was influenced by two dominant weather systems, the South Asia High and the Western Pacific Subtropical High (Fig S4). The former was positioned at the upper troposphere (around 200 hpa), with its center located in the northern part of the SCB. Meanwhile, the latter was situated within the



troposphere (lower than the former), with its high-pressure ridge extending
from east to west, covering the entire SCB region. Under the influence of
the two major high-pressure systems, the SCB experienced subsidence
airflows, resulting in a stationary atmospheric structure. According to the
synoptic flows (Fig S4), it could be seen the prevailing wind was
southeastward, and the wind speed gradually decreased from east to west,
implying that Chengdu was more stationary than Chongqing.

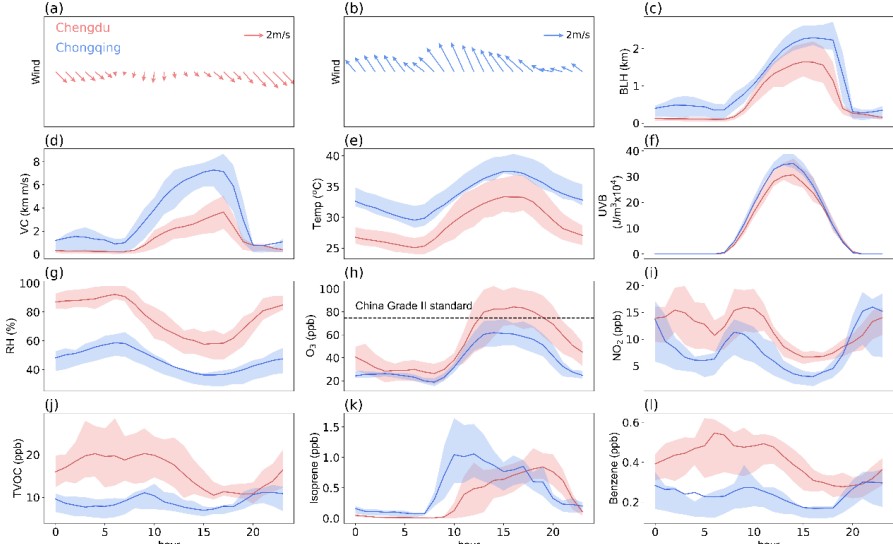


Fig 2 Diurnal variation of meteorological parameters (including winds, boundary layer
height (BLH), temperature (temp), relative humidity (RH) and ultraviolet radiation (UVB))
and air pollutants ($O_3$, $NO_2$, total volatile organic carbons (TVOC) and isoprene) in Chengdu
and Chongqing, respectively
Furthermore, we compared the averaged diurnal variations of the in-situ
measured meteorological parameters and air pollutants (Fig 2). Consistent
with the analysis of weather patterns, Chongqing was influenced by the
southeast winds (3.1 m/s), while Chengdu was more stagnant with lighter
wind speed (1.4 m/s) (Fig 2a-b). In addition, the boundary layer height
(BLH) was also significantly higher in Chongqing (Fig 2c). A simple
calculation of the ventilation coefficient (VC) with wind speed and BLH
indicated that Chongqing (VC=3.34 km·m/s) had better ventilation
conditions compared to Chengdu (VC=1.24 km·m/s, Fig 2d). It could be
inferred that, influenced by lighter winds and lower BLH, air pollutants in
Chengdu were more easily trapped and accumulated. Both cities displayed



typical meteorological features of a heatwave conducive to photochemical
pollution, characterized by elevated temperatures, intense solar radiation,
and low relative humidity (Fig 2e-2g). Among these factors, both
temperature and solar radiation in Chongqing were higher compared to
those in Chengdu, suggesting that the conditions in Chongqing were more
conducive to photochemical $O_3$ reactions. However, the degree of $O_3$
pollution was quite the opposite as previously mentioned (Fig 1a and Fig
2h). We conducted further investigation into the diurnal variation of the
precursors. Two distinct peaks in $NO_2$ levels were identifiable, with one
occurring in the morning and the other appearing during night (Fig 2i). The
morning peaks were likely influenced by vehicular emissions during rush
hours. The night peaks were possibly caused by the $NO_x$ titration effect.
Moreover, the levels of total VOCs (TVOC) were much higher in Chengdu
than those in Chongqing (Fig 2j). Considering the different degrees of $NO_2$
and TVOC concentrations in Chengdu and Chongqing, it could be inferred
that there might be differences in the $O_3$ formation mechanism between the
two cities. Indeed, the diurnal variation of isoprene, a highly active VOCs
compound, showed distinct differences (Fig 2k). The observed data in
Chongqing showed a notable afternoon peak, whereas in Chengdu, the
peak appeared exclusively between 17:00 and 20:00. Usually, isoprene,
mainly emitted by vegetation, is sensitive to ambient temperature and solar
radiation and peaks at noon time. There might be some potential
explanations. Firstly, the isoprene peak between 17:00 and 20:00 in
Chengdu could be attributed to other sources, such as vehicular emissions.
However, this possibility was ruled out after examining the diurnal
variation of benzene (Fig 2l). As a marker of anthropogenic vehicular
emissions, benzene did not exhibit any peaks between 17:00 and 20:00.
The second possibility was that the atmospheric oxidizing capacity in
Chengdu was more robust than in Chongqing, leading to the rapid
photochemical consumption of isoprene emitted by vegetation. This
hypothesis was supported by the diurnal variations in $O_3$ levels, which were
notably elevated in the afternoon, implying of a stronger atmospheric
oxidizing capacity. The instrument-detected of isoprene was indicative of
its "aged" state, implying the rapid photochemical consumption due to both



the atmospheric oxidizing capacity and the inherent reactivity of isoprene
itself. Furthermore, a distinct decrease of BLH between 17:00 and 20:00
was also a possible reason causing the isoprene peak of Chengdu in the late
afternoon.
Subsequently, we employed a machine learning method, the Stepwise
Regression Analysis, to quantify the impact of diverse meteorological
parameters and precursor concentrations on $O_3$ levels. In both cities, the
significance of T2 and SSR, along with ACCO3 and ACCNO2, took
precedence. This indicates that meteorological conditions characterized by
high temperatures, intense solar radiation, and the presence of overnight
accumulative pollutants played a pivotal role in $O_3$ concentration,
especially during heatwaves. The distinction between the two cities lied in
the significance of atmospheric dispersion capacities represented by the
variations in winds and BLH. The study revealed that winds, including both
horizontal winds (U10 and V10) and vertical wind (W), along with BLH,
had positive effects in elevating $O_3$ levels in Chengdu. Conversely, they
predominantly had negative effects, resulting in a decrease in $O_3$ levels, in
Chongqing. These findings align with the diurnal analysis, which indicated
that Chengdu experienced lighter winds and lower BLH. The poor
ventilation conditions facilitated the accumulation of air pollutants,
contributing to the increase in $O_3$ levels. In contrast, the ventilation
condition in Chongqing was conducive to reduce $O_3$ concentrations.
Combined with the aforementioned analysis of diurnal patterns, it could be
inferred that Chengdu was more constrained by local emissions, while
Chongqing was more susceptible to regional transport influences (further
discussed in Section 3.3).





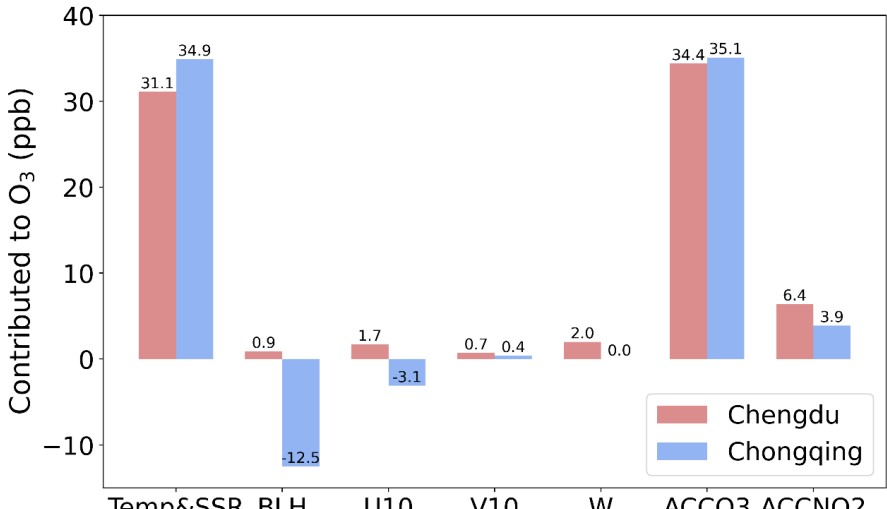

Fig 3 Contribution of multi-factors influencing $O_3$ concentrations in Chengdu and Chongqing, respectively

### 3.2 Difference in heatwave-intensified BVOCs emissions and their impact on $O_3$ formation

In addition to the influence of meteorological factors under heatwave conditions, the precursors also play important roles in contributing $O_3$ concentrations. Therefore, we utilized the OBM model to compute and identify the primary VOCs components that exerted a substantial influence on $O_3$ levels. Here, we introduced the RIR values that could reflect the importance of a given species to $O_3$ concentrations. As Fig 4 shows, Alkenes and aromatic hydrocarbons were the principal VOCs components influencing $O_3$ levels in both cities. In Chengdu, the most influential VOCs species on $O_3$ concentrations included isoprene, m-xylene, trans-2-butene, o-xylene, cis-2-butene, toluene, ethene, 1-hexene, 1,2,4-trimethylbenzene, and 1,2,3-trimethylbenzene. Similarly, in Chongqing, the primary VOCs contributors to $O_3$ levels were isoprene, m-xylene, trans-2-butene, cis-2-butene, o-xylene, 1,2,4-trimethylbenzene, trans-2-pentene, propane, cis-2-pentene, and toluene. According to the results, both Chengdu and Chongqing should prioritize the regulation of alkenes and aromatic hydrocarbons from sources like vehicular emissions and solvent usage. Besides, the results clearly highlight isoprene as the dominant VOCs species impacting $O_3$ levels. The characteristics of the heatwave were high



temperature, intense solar radiation and dry air condition. These
meteorological factors significantly enhanced the emission of BVOCs
from vegetation, indicating the notable role of heatwave-triggered natural
emissions in the secondary $O_3$ pollution.

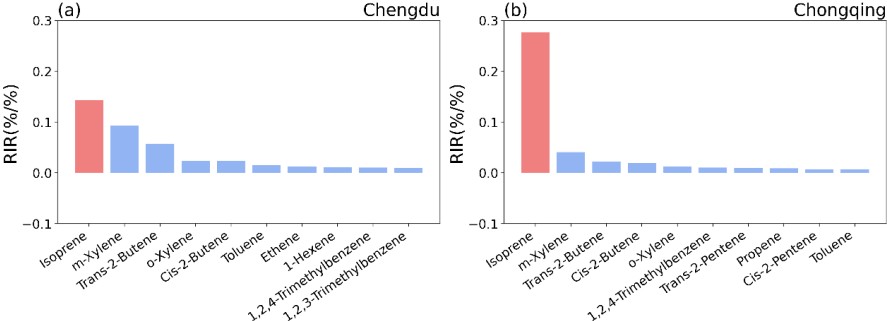

Fig 4 OBM calculated the top 10 VOCs species with the highest RIR values in (a) Chengdu
and (b) Chongqing.

We further examined the relationship between isoprene, temperature, and
$O_3$ using observational data. In order to expand the sample size, we
gathered a dataset corresponding to the daily maxima $O_3$ values recorded
during the months of July and August in 2022. In Chengdu, the variations
of isoprene and temperature basically showed an increasing trend,
indicating that higher isoprene concentrations were associated with higher
temperatures, which in turn coincided with elevated $O_3$ levels (Fig 5a). In
Chongqing, the concentration of isoprene initially increased with rising
temperatures. However, when the temperature surpassed approximately
40°C, the isoprene concentration started to decrease with further
temperature elevation (Fig 5b). Notably, the peak values of $O_3$
corresponded closely to the high values of isoprene, occurring at
temperatures ~ 38°C to ~ 42°C. According to recent studies, isoprene
emissions increase with rising temperatures, and even under high-
temperature conditions when vegetation closes stomata, due to the indirect
impact of elevated leaf temperature, it decreases only under extreme high-
temperature drought conditions because of the inhibition of substrate
supply (Potosnak et al., 2014; Wang et al., 2022a). Here, the variation of
isoprene with temperature in Chengdu and Chongqing illustrates these two
distinctions though the isoprene concentration being observed was "aged".



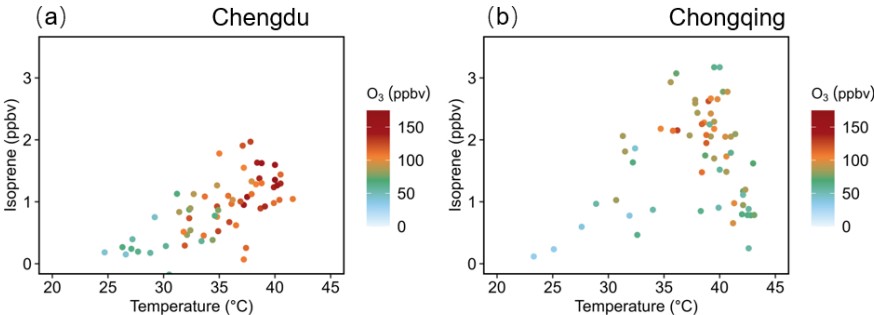

Fig 5 Scatter plots of observed isoprene, temperature and $O_3$ in (a) Chengdu and (b) Chongqing. The data were collected corresponded to daily maxima $O_3$ concentrations from July 2022 to August 2022.

We utilized the theoretical calculation from MEGAN model to quantify the disparities in isoprene emissions between the two cities. Considering the varying administrative areas of Chengdu (14,378 km²) and Chongqing (82,339 km²), comparing the total isoprene emissions might not be appropriate. Instead, we quantified the emissions per unit grid area (9×9 km) for both locations (Fig 6). It can be observed that the isoprene emissions in Chongqing were higher than those in Chengdu (nearly twice as much). In particular, under the influence of heatwaves, the isoprene emissions in Chongqing and Chengdu increased by 41.1% and 22.2%, respectively. The significant role of heatwave-intensified BVOCs emissions was expected to aggravate $O_3$ pollution in Chengdu and Chongqing. With the aid of CMAQ-ISAM simulation, we proceeded to quantify the distinct impacts of anthropogenic emissions and BVOCs emissions on $O_3$ concentrations. The findings indicated that at 13:00 (local time), when photochemical reactions were most intense, anthropogenic emissions contributed to 50.6 ppbv and BVOCs emissions contributed to 33.3 ppbv in Chengdu. In comparison, anthropogenic emissions and BVOCs emissions contributed to 31.3 ppbv and 20.6 ppbv in Chongqing, respectively. Interestingly, despite higher BVOCs emissions in Chongqing compared to Chengdu, the contribution of BVOCs to $O_3$ levels was actually smaller in Chongqing than in Chengdu. This implies that there were differences in the $O_3$-$NO_x$-VOCs response mechanisms between the two cities.



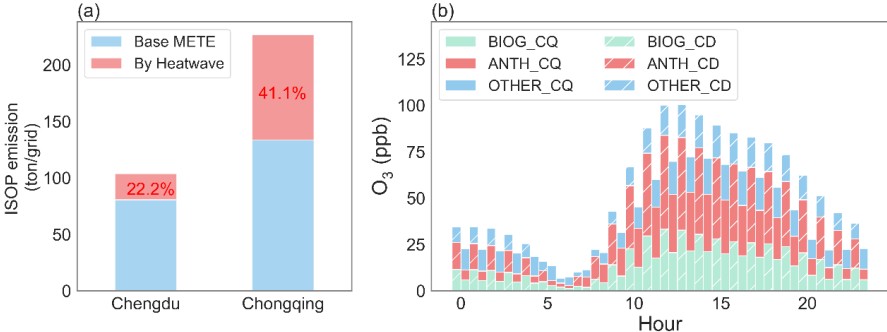

Fig 6 (a) Meteorology driven ISOP emission between Chengdu and Chongqing, respectively.;
(b) Averaged source contributions (by emissions) to diurnal $O_3$ levels in Chengdu (CD) and
Chongqing (CQ), respectively. BIOG, ANTH and OTHER refer to contributions from the
biogenic, the anthropogenic and the others.

Herein, we conducted CMAQ-DDM simulations to investigate the nonlinear relationship between $O_3$ and its precursors. Indeed, the $O_3$-$NO_x$-VOCs sensitivity response mechanisms in Chengdu and Chongqing were of difference (Fig 7 a-b). On the one hand, the Chengdu region demonstrated a greater sensitivity (first-order sensitivity coefficients) to VOCs in comparison to Chongqing. Specifically, in certain urban grids within Chengdu, the sensitivity coefficient exceeded 10 ppbv, while the highest sensitivity in Chongqing was only ~ 3 ppbv. On the other hand, Chongqing generally exhibited higher sensitivity to $NO_x$, except for quite limited urban cores. In contrast, the eastern areas of Chengdu, particularly its urban cores, displayed low sensitivity to $NO_x$. Furthermore, by taking both the first-order sensitivity coefficient and the 2nd-order sensitivity coefficient into account, we constructed the $O_3$ isopleth for both cities during the month of August (Fig 7 c-d). It was evident that Chengdu was situated in a VOCs-limited regime, while Chongqing was operating within a mixed-limited regime. These simulated results agree with the satellite diagnosed $O_3$ formation sensitivity (obtained through the ratio of HCHO and $NO_2$), confirming again the good modeling performance (Fig S5). The results implied that a temporary decrease in $NO_x$ emissions in Chengdu would result in an increase in $O_3$ concentrations, whereas reducing VOCs emissions could potentially lower $O_3$ pollution. This finding could partially explain the increasing trend of $O_3$ concentrations in Chengdu Plain during





the past as the previous emission control measures were mainly targeted to
$NO_x$ emissions (driven by the need to control acid rain and $PM_{2.5}$ pollution,
successively). In Chongqing, differently, a reduction in either $NO_x$
emissions or VOCs emissions could contribute to alleviating $O_3$ pollution.
The disparity in $O_3$-$NO_x$-VOCs sensitivity between the two cities could
also elucidate the reason why Chongqing, despite its higher BVOCs
emissions, exhibits a lower contribution to $O_3$ levels. Considering the
varying regional sensitivities in $O_3$-$NO_x$-VOCs formation, it is advisable to
implement precise emission reduction strategies tailored to the unique
sensitivities of each city for effective pollution prevention and control. This
approach stands in contrast to a uniform solution that may not suit all
contexts. For example, in Chengdu, the previously nationally implemented
strategy, which prioritized $NO_x$-focused control, might ultimately lead to
$O_3$ reduction through substantial $NO_x$ reductions. However, this approach
would initially enter into a phase characterized by relatively high $O_3$
concentrations (positioned within the transitional regime based on the $O_3$
isopleth), posing environmental risks. Instead, a strategy centered on VOCs
control alongside simultaneous $NO_x$ control could bypass the "high-$O_3$"
phase and align with the need to address both $O_3$ and $PM_{2.5}$ pollution.

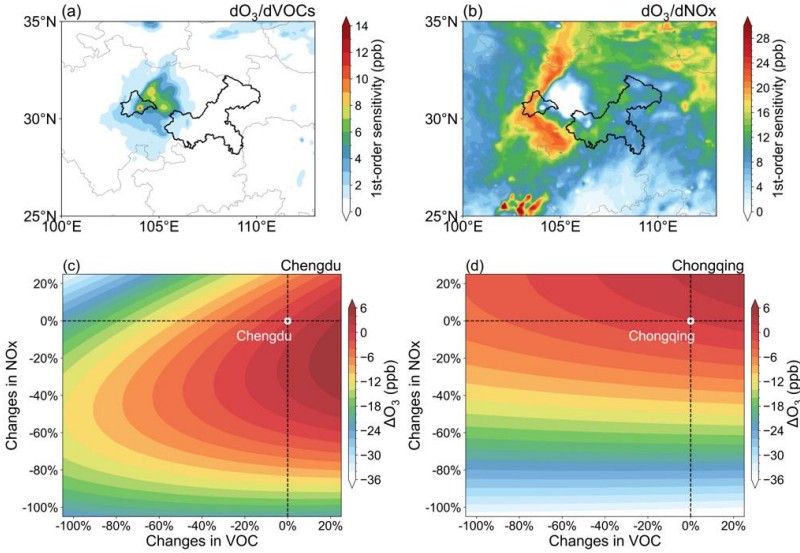


Fig 7 Spatial distribution of daytime first-order sensitivity coefficients to (a) VOCs and (b)
$NO_x$; $O_3$ isopleth plots in (c) Chengdu and (d) Chongqing



## 3.3 Regional divergence of source region contribution

Surface-level $O_3$ is influenced not only by photochemical reactions but also by regional transport. In this section, we mainly focus on the disparities in the impact of regional transport on $O_3$ between Chengdu and Chongqing. Fig 8 demonstrates the LPDM simulated 72h backward retroplumes influencing Chengdu and Chongqing. In general, Chengdu was primarily influenced by local air masses, encompassing areas such as Chengdu city and the eastern parts of the SCB. Relatively fewer air masses originated from cross-province transport in the southeast direction. Differently, Chongqing showed a situation to be more susceptible to cross-regional transport influences. The dominant air masses in Chongqing not only originated locally but also experienced cross-province transport from the southeast, influenced by the regions such as Guizhou and Guangxi with cleaner ambient air masses. The results from the LPDM simulations closely aligned with the Stepwise Regression Analysis as showed in Section 3.1. Besides, we also adopted the in-situ measured data by comparing the ratio of m, p-xylene and ethylbenzene. Given that m, p-xylene is more reactive than ethylbenzene, their ratios typically decrease due to photochemical reactions that take place during the transport of air masses. As shown in Fig S6, the ratio was much lower in Chongqing (1.04 ppbv ppbv$^{-1}$), indicating the presence of "aged" air masses being monitored. Conversely, a higher ratio (3.11 ppbv ppbv$^{-1}$) in Chengdu indicated the prevalence of "fresh" air masses likely originating from local emissions. The discovery reaffirmed that Chongqing exhibited superior ventilation conditions compared to Chengdu. This inference suggests that Chongqing's enhanced dispersion capacity played a pivotal role in significantly reducing its O3 concentrations during the severe heatwave period.



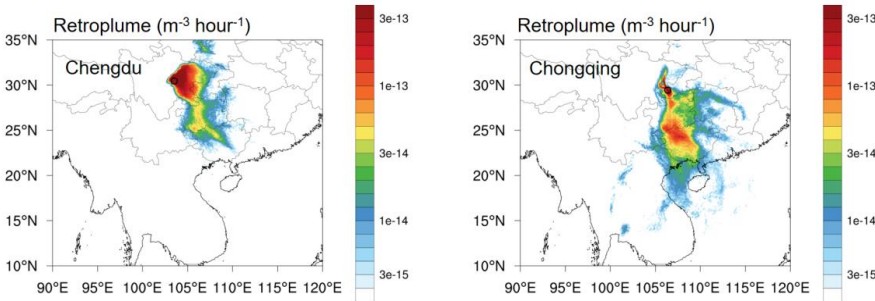

Fig 8 Comparison of 72h retroplume (footprint residence time) showing transport pathways of
air masses arriving at Chengdu and Chongqing in August 2022

Further, we employed the CMAQ-ISMA modeling system to quantify the source region's contribution to Chengdu and Chongqing (Fig 9). In this study, we divided the study area into eight major regions, namely Chengdu Plain (CD Plain), Chongqing (CQ), South Sichuan (South SC), Northeast Sichuan (Northeast SC), Northwest Region (Northwest), Southwest Region (Southwest), Northeast Region (Northeast), and Southeast Region (Southeast)(Fig S2). Generally, the regions like CD Plain, CQ, Northeast SC, South SC were distributed within the SCB region, and could be regarded as the local regions. On the other hand, regions like Northwest, Southwest, Northeast, and Southeast were situated outside the SCB and air masses originating from these regions were considered to be a result of regional transport. As Fig 9 shows, Chengdu was mainly affected by local regions, contributed to 46.8%. This implied that local emissions within the SCB were a significant contributor to the excessive $O_3$ levels in Chengdu. In contrast, the influence of the local region on $O_3$ levels in Chongqing was only 18.3%. Instead, the contribution outside the basin almost reached 50%, indicating that Chongqing was more susceptible to the influence of cross-regional transport. This difference demonstrates that even the two major core cities located within the SCB exhibit distinct source contribution characteristics.





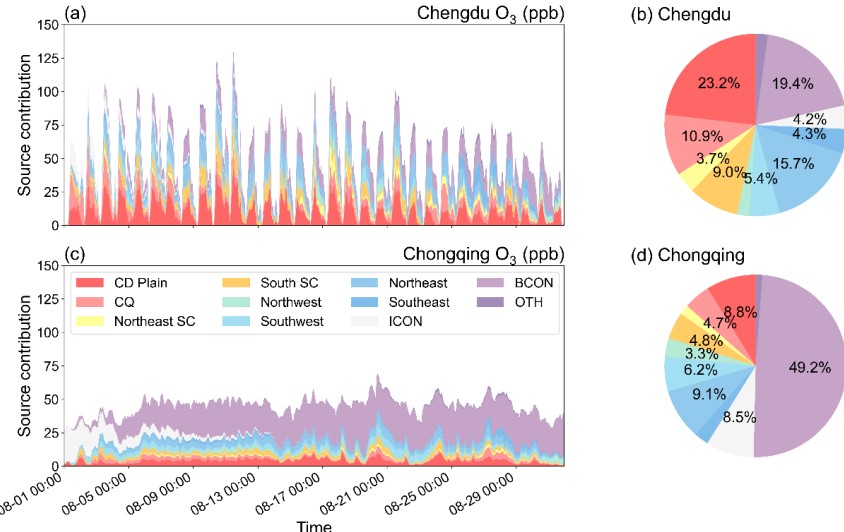

566

Fig 9 Source region's contribution to $O_3$ levels in Chengdu (a) Time series of $O_3$ contributions
from each region. (b) Pie charts illustrating the percentage contributions of each region. (c)
and (d) same as (a) and (b), but in Chongqing

Given that ambient $O_3$ concentrations are the integrated results of multiple
processes, encompassing photochemical formation, deposition, and
transport, we employed the Integrated Process Rate (IPR) tool within the
CMAQ model to analyze the contributions of individual physical and
chemical processes to $O_3$ levels. Here, we compared the contributions of
different processes to $O_3$ during the peak period of heightened
photochemical reactions at 14:00 in the afternoon. As Fig. 10 shows, the
process analysis results reveal distinct differences between the two cities.
Specifically, in Chengdu, photochemical reactions took the lead in
escalating $O_3$ levels (reaching 49.9 ppbv). This could be attributed to a
combination of factors. On one hand, being limited to the local air masses,
pollutants got accumulated and resulted in the increment of the
atmospheric oxidizing capacity. On the other hand, under the influence of
conducive meteorological conditions during heatwaves, the vigorous
photochemical formation of $O_3$ was substantially enhanced, resulting in
notable $O_3$ concentration increments. Compared to Chengdu, the
contribution of photochemistry to $O_3$ in Chongqing was nearly half (29.2
ppbv). While both photochemical reactions and regional transport





positively affected O$_3$ levels in Chongqing, the overall net accumulation of
O$_3$ was notably lower in this city.

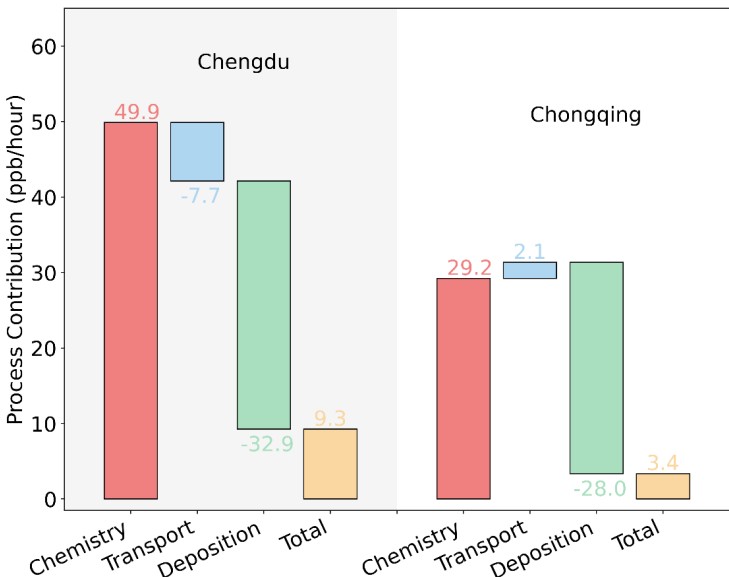


Fig 10 Averaged contributions of different process to O$_3$ concentrations at noon time (14:00)
in Chengdu and Chongqing

## 4. Conclusion and implication

The unprecedented heatwave of August 2022 brought about significant
divergence in O$_3$ levels between Chengdu and Chongqing, with exceeded
levels of O$_3$ appeared in the western SCB (Chengdu) but relatively lower
concentrations in the eastern basin (Chongqing). Meteorological and
precursor factors were assessed using a machine learning method,
spotlighting high temperatures, intensive solar radiation, and overnight
accumulative pollutants as key contributors to O$_3$ concentration. The
interplay of isoprene, temperature, and O$_3$, alongside MEGAN calculations,
underscored the intensified BVOCs emissions during heatwaves,
highlighting the important role of meteorology-induced natural emissions.
Interestingly, BVOCs emissions in Chongqing were nearly twice those in
Chengdu; however, their contributions to O$_3$ concentrations were subdued.
This discrepancy was attributed to the distinct responses of O$_3$-NO$_x$-VOCs
sensitivity mechanisms. Chengdu exhibited sensitivity to VOCs, while




Chongqing displayed a transitional sensitivity regime. Considering that China's previous emission reduction strategies have primarily focused on a nationwide $NO_x$ reduction (driven by the need to control $PM_{2.5}$ pollution), it is important to recognize that a short-term reduction in $NO_x$ can lead to an $O_3$ rebound in regions like Chengdu Plain. To achieve more precise pollution control, a strategy that combines VOCs as the primary focus with concurrent $NO_x$ reductions would be more appropriate. In addition, the investigation into source region contributions revealed varying impacts of regional transport, even within the same basin. Chongqing was significantly influenced by cross-regional transport, whereas Chengdu was predominantly affected by local emissions.

These findings illuminate the complex interplay of meteorology, natural emissions, and anthropogenic sources during heatwaves, guiding the necessity of targeted pollution control measures. It is imperative to adopt emission control strategies that are customized according to regional or even local conditions, rather than enforcing uniform measures for the entire region. Given that $O_3$ pollution is not solely an in-situ problem but rather a regional issue, this concept extends beyond the SCB and is applicable to other urban clusters, such as the Beijing-Tianjin-Hebei region, the Yangtze River Delta region, the Pearl River Delta region, and developed regions in other countries. Future efforts are suggested to focus on regional coordinated and balanced control measures.

**Author Contributions**

F.Y. and N.W. designed the research. N.W. wrote the manuscript. N.W., D.Y., C.D., and M.H. contributed to the interpretation of the results. All the authors provided critical feedback and helped to improve the manuscript.

**Competing Interests**

The authors declare that they have no known competing financial interests or personal relationships that could have appeared to influence the work.

**Acknowledgement**

This study is supported by the National Natural Science Foundation of China (No. 42175124 and No. 22276128), Science and Technology Department of Sichuan Province (23YFS0383), the Guangdong Basic and Applied Basic Research Foundation (Grant No. 2022A1515011753), the




Fundamental Research Funds for the Central Universities (Grant No. YJ202313), and the Young Talent Support Project of Guangzhou Association for Science and Technology (Grant No. QT-2023-048). The authors also thank the Tsinghua University for compiling and sharing the MEIC emission inventory.

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
