# Peer review of "Spatial Disparities of Ozone Pollution in the Sichuan Basin Spurred by an Extreme Heatwave"

_EGUsphere, 2023_

## Referee Comment (RC2)

The unprecedented heatwave of August 2022 brought about national wide $O_3$ pollution in China. This study targeted the heatwave month of August, 2022 in the air pollution basin SCB in Southwest China presenting the detailed investigation on the spatial disparity of $O_3$ pollution between two major urban areas Chongqing and Chengdu with pollution levels of $O_3$ in the western SCB (Chengdu) but relatively lower concentrations in the eastern basin (Chongqing). Meteorological and precursor factors were assessed with observation, modeling study and machine learning methods, spotlighting high temperatures, intensive solar radiation, and overnight accumulative pollutants as key contributors to $O_3$ concentration, revealing the important role of meteorology-induced natural emissions and meteorological changes. It is suggested to be accepted this paper for ACP publication in the ACP after the major revisions:

**Major comments:**

As mentioned in the discussions, if the cross-regional transport predominantly influenced Chongqing (contributing ~80%), the local photochemical $O_3$ production contributed merely about 20 % to $O_3$ variations during August 2022 in Chongqing, where the prevailing southerly drived the transport of poor-$O_3$ air masses from the Yungui Plateau,because Chongqing is immediately adjacent to the Yungui Plateau in the south. Please give the detailed discussions and highlighted the results on the regional transport importance in the spatial disparity of O3 pollution between Chongqing and Chengdu in the following respects of changing emission and meteorology:

a) the lower $O_3$ in Chongqing was dominated by the the transport of poor-$O_3$ air masses from the Yungui Plateau with the less contribution of photochemical O3 production from local and regional transported precursors.

b) the cross-regional transport of $O_3$-precursors from the Yungui Plateau with low anthropogenic emissions and high natural emissions. It is possible to identify the local and non-local $O_3$-precursors with their relative contribution in Chongqing.

c) the spatial disparity of $O_3$ pollution between Chongqing and Chengdu was decided by the changes in regional transport of $O_3$ and its precursors respectively with high and low contribution of non-local $O_3$-precursors to the two urban areas.

d) Please compare the 33-yr (1990-2022) averages of air temperature, relative humidity, wind speed

and direction over August between Chengdu and Chengdu with the anomalies in August,2022. Please clarify that the anomalies of high air temperature and low relative humidity are connected with the strong-southerly-driven cross-regional transport of cool and dry air masses from the Yungui Plateau, which could suppress the photochemical O3 production in Chongqing.

**Specific comments:**

1) Observational data showed that Chengdu experienced a consecutive 17-day period of $O_3$ exceedance,Did the extreme heatwave last the consecutive 17-day period in Chengdu? How long was the extreme heatwave in Chongqing? As the study targeted the month of August, 2022 (not a heatwave), please modify the title of manuscript for the study.

2) The first paragraph in sect. 1 Introduction is too long with the disorder presentation. Please restructure this paragraph in 2-3 paragraph.

3) Please mark the location of Tibetan Plateau and Yungui Plateau and indicate the meaning of color contours in Fig. 1a. Please change Fig. 1b with the 33-yr averages of air temperature over August between Chengdu and Chengdu with the anomalies in August over 1990-2022 based on the site observations.

4) Please carefully check the caption of Fig. 2 against the Figure, give a correct caption (vc, wind, averages over August 2022).

5) Fig. 1g presented the significant differences in RH. Please add the discussions on role of RH in the spatial disparity of O3 pollution between Chongqing and Chengdu.

6) Please remove the discussions (lines 302-313) on the synoptic system. 200hPa and 500hPa are too high to discuss the stationary and unstable atmosphere for air quality change over the SCB.

7) Please thoroughly check the English presentations in the text and all the Figs. with the clear and captions.

---

## Author Comment (AC1)

**Response to Reviewer 1**

**General Comments:**

**The study investigates the mechanism of regional ozone (O3) pollution disparities in the Sichuan Basin during a severe heatwave event. Chengdu experienced a 17-day O3 exceedance, while the O3 levels in Chongqing were below the national standard. It was found that high temperature and solar radiation intensity, as well as the accumulation of pollutants, were the key contributors to the O3 episodes in Chengdu. Meanwhile, model simulations revealed that the O3 formation regime is VOC-sensitive in Chengdu and transitional in Chongqing. As a result, the biogenic volatile organic compounds (BVOCs), especially isoprene, played a significant role in O3 formation in Chengdu, while contributing less in Chongqing, despite its higher emissions. Besides, in comparison with the strong local O3 formation in Chengdu, regional transport influenced the O3 levels in Chongqing predominantly, highlighting the need for targeted pollution control measures on the regional scale. Overall, this manuscript is well-written and of interesting scientific value. However, there are some minor issues and questions that require attention before publication. Therefore, I would recommend acceptance of this manuscript after the following issues are addressed.**

**General response to the reviewer 1:** Thank you very much for your valuable comments and suggestions. Your positive comments/suggestions have motivated us to improve the manuscript. Now, we have carefully revised the manuscript based on all your questions/suggestions, and hope the correction will meet with approval. We have marked the revised sentences in red color in the manuscript. Below is the point-to-point response.

**In Lines 138-140: More specificity is needed in describing the locations and surroundings of the sampling sites. The authors could also mark their locations in Fig 1 to clarify.**

**Reply:** Thanks for the suggestion. The following descriptions were added in the manuscript to introduce the sampling sites, "Data of atmospheric compositions, including $O_3$, $NO_x$ (NO and $NO_2$), CO, $SO_2$, VOCs components and meteorological parameters were collected from two in-situ observational sites. The Chengdu sampling site was located on the rooftop super monitoring station of the Chengdu Environmental Science Academy in Qingyang District, Chengdu (30.65°N, 106.49°E), while the Chongqing sampling site was situated on the rooftop research observation station of Longshan Primary School in Yubei District, Chongqing (29.75°N, 106.46°E). Both sites were situated in mixed-use areas encompassing traffic arteries, commercial, and residential zones, serving as representative locations for assessing urban air quality."

In addition, their locations were also marked in Fig 1 as suggested.

[Figure]

Fig 1 (a) Geographical distribution of Sichuan Basin with scattered averaged monthly MDA8 $O_3$ concentrations (data obtained from Ministry of Ecology and Environment of China). The black lines highlight the administrative border of Chengdu and Chongqing, respectively. The blue star shows the locations of the supersites. (b) Historical monthly averaged daily-maximum air temperature (August) variation of the SCB from 1990 to 2022. The red bar highlights the extreme hot temperature in 2022.

**In Lines 146-147: Online GC instruments were used to measure VOCs. What about the quality control of these instruments, such as the limit of detection, precision, and accuracy?**

**Reply:** The online GC instruments was GC955-611/811(by Synspec). The instrument targeted the VOCs species designated as photochemical precursors by the US Environmental Protection Agency (EPA). The gas standards used were identical to those employed by the US EPA Photochemical Assessment Monitoring Stations (PAMS). The calibration curves and detection limits of VOCs species were summarized in Table R1.

Table R1* The calibration curves and detection limits of VOC species

| Target compound | Calibration curve | Correlation coefficient | Detection limit (ppbv) |
|---|---|---|---|
| Ethene | $y = 1.0188x + 0.2659$ | 0.997 | 0.07 |
| Acetylene | $y = 1.0409x + 0.1756$ | 0.998 | 0.08 |
| Ethane | $y = 1.0162x + 0.2891$ | 0.997 | 0.08 |
| Propene | $y = 0.9959x + 0.1506$ | 0.999 | 0.07 |
| Propane | $y = 0.9824x + 0.2082$ | 0.998 | 0.09 |
| i-Butane | $y = 0.9753x + 0.3785$ | 0.994 | 0.05 |
| 1-Butene | $y = 0.9587x + 0.3641$ | 0.994 | 0.06 |
| n-Butane | $y = 0.9776x + 0.3718$ | 0.994 | 0.05 |
| trans-2-Butene | $y = 0.9746x + 0.2747$ | 0.997 | 0.05 |
| cis-2-Butene | $y = 0.9834x + 0.1606$ | 0.999 | 0.06 |
| i-Pentane | $y = 0.9753x + 0.2135$ | 0.998 | 0.07 |
| 1-Pentene | $y = 0.919x + 0.1626$ | 0.998 | 0.05 |
| n-Pentane | $y = 0.9557x + 0.2038$ | 0.984 | 0.07 |
| Isoprene | $y = 1.0304x + 0.1653$ | 0.998 | 0.07 |
| trans-2-pentene | $y = 0.9753x + 0.2135$ | 0.998 | 0.07 |
| cis-2-pentene | $y = 0.9557x + 0.2038$ | 0.984 | 0.07 |
| 2,2-Dimethylbutane | $y = 0.9731x + 0.1971$ | 0.998 | 0.07 |
| Cyclopentane | $y = 0.9993x + 0.1412$ | 0.997 | 0.06 |
| 2,3-Dimethylbutane | $y = 0.919x + 0.1626$ | 0.999 | 0.07 |
| 2-Methylpentane | $y = 0.9557x + 0.2038$ | 0.984 | 0.07 |
| 3-Methylpentane | $y = 0.9753x + 0.2135$ | 0.998 | 0.07 |
| 1-Hexene | $y = 0.9700x + 0.3300$ | 0.995 | 0.05 |
| n-Hexane | $y = 0.9915x + 0.2626$ | 0.997 | 0.06 |
| Methylcyclopentane | $y = 0.9749x + 0.1832$ | 0.999 | 0.07 |
| 2,4-Dimethylpentane | $y = 0.9993x + 0.1412$ | 0.999 | 0.05 |
| Benzene | $y = 0.9753x + 0.2835$ | 0.997 | 0.06 |
| Cyclohexane | $y = 0.9841x + 0.2744$ | 0.997 | 0.07 |
| 2-methylhexane | $y = 0.9744x + 0.2979$ | 0.996 | 0.05 |
| 2,3-dimethylpentane | $y = 0.9779x + 0.2953$ | 0.997 | 0.05 |
| 3-methylhexane | $y = 0.9735x + 0.3374$ | 0.995 | 0.05 |
| 2,2,4-trimethylpentane | $y = 0.9696x + 0.3947$ | 0.994 | 0.05 |
| n-Heptane | $y = 0.9678x + 0.3635$ | 0.994 | 0.05 |
| Methylcyclohexane | $y = 0.9819x + 0.3629$ | 0.995 | 0.05 |
| 2,3,4-trimethylpentane | $y = 0.9691x + 0.3994$ | 0.994 | 0.04 |
| Toluene | $y = 0.9696x + 0.3397$ | 0.995 | 0.05 |
| 2-methylheptane | $y = 0.9603x + 0.4835$ | 0.990 | 0.04 |
| 3-methylheptane | $y = 0.9625x + 0.4550$ | 0.991 | 0.04 |
| n-Octane | $y = 0.9524x + 0.5082$ | 0.989 | 0.04 |
| Ethylbenzene | $y = 0.9629x + 0.4253$ | 0.992 | 0.04 |
| m, p- Xylenes | $y = 0.9541x + 0.5844$ | 0.986 | 0.03 |
| Styrene | $y = 0.9524x + 0.4132$ | 0.991 | 0.04 |
| o-Xylene | $y = 0.9515x + 0.4926$ | 0.989 | 0.04 |
| n-Nonane | $y = 0.9878x + 0.1635$ | 0.998 | 0.04 |
| i-Propylbenzene | $y = 0.9418x + 0.5162$ | 0.986 | 0.04 |
| n-Propylbenzene | $y = 0.9426x + 0.5468$ | 0.986 | 0.04 |
| m-Ethyltoluene | $y = 0.9532x + 0.4838$ | 0.989 | 0.04 |
| p-Ethyltoluene | $y = 0.9554x + 0.3953$ | 0.992 | 0.04 |
| 1,3,5-Trimethylbenzene | $y = 0.951x + 0.4724$ | 0.989 | 0.04 |
| o-Ethyltoluene | $y = 0.9784x + 0.0956$ | 0.999 | 0.04 |
| 1,2,4-trimethylbenzene | $y = 0.9563x + 0.4509$ | 0.991 | 0.03 |
| n-Decane | $y = 0.9651x + 0.3068$ | 0.995 | 0.04 |
| 1,2,3-trimethylbenzene | $y = 0.9537x + 0.3191$ | 0.993 | 0.04 |
| m-Diethylbenzene | $y = 0.9541x + 0.4494$ | 0.991 | 0.04 |
| p-Diethylbenzene | $y = 0.9607x + 0.3788$ | 0.993 | 0.04 |
| n-Undecane | $y = 0.9519x + 0.3329$ | 0.992 | 0.04 |
| n-Dodecane | $y = 0.9890x + 0.2711$ | 0.993 | 0.05 |

**In Lines 176-178: Please provide the input details for the machine learning simulations. For example, which set of data was used to train the model, and which set of data was used for validation? In addition, in Line 177, please**

**clarify the R and P values. Is it the Pearson correlation coefficient or coefficients of determination and is the p-value calculated from one- or two-tailed t-test?**

**Reply:** Thanks for the question. In our study, we didn't use machine learning to do prediction. We took advantage of Stepwise Regression Analysis to explain how meteorological factors impact $O_3$ concentrations. In detail, we constructed a multivariate linear regression equation to model $O_3$ concentration. Meteorological parameters were obtained from the fifth generation of the European Centre for Medium-Range Weather Forecasts atmospheric reanalysis (ERA5). The selected parameters included 10m u-component of wind (U10), 10m v-component of wind (V10), vertical wind (w), boundary layer height (BLH), 2m temperature (T2) and surface solar radiation (SSR). Additionally, we also incorporated previous night accumulative air pollutants, such as $O_3$ (ACCO3) and $NO_2$ (ACCNO2), as input parameters to investigate the impact of pollutants being overnight accumulated on $O_3$ levels. The machine learning-simulated $O_3$ concentrations were then validated against observations, revealing a robust correlation (Pearson correlation coefficient (R) > 0.91, p-Value (from two-tailed t-test) < 0.01) between them (Fig S1).

**In Line 191: JNO2 were measured during the sampling period, while the authors did not introduce the instrument for JNO2 measurements. Please provide this information in Section 2.1.**

**Reply:** Thanks for the careful review. We have added "The photolysis rate of $NO_2$ ($J_{NO2}$ value) were measured by Ultra-fast CCD-Detector Spectrometer (UF-CCD, MetCon, Germany)" in Section 2.1

**In Lines 216-220: Please explain how the backward trajectory of 3000 particulates was.**

**Reply:** In this study, the LPDM simulation was conducted with the aim to understand the potential source region impacting Chengdu and Chonging. We released 3000 particulates as tracers over the site to investigate their 72-hour backward movement. The calculation of the backward movement was using the HTSPLIT model. Generally, the calculation is a hybrid between the Lagrangian approach, using a moving frame of reference for the advection and diffusion calculations as the trajectories or air parcels move from their initial location, and the Eulerian methodology, which uses a fixed three-dimensional grid as a frame of reference to compute pollutant air concentrations. The introduction of the LPDM has been presented in Section 2.4.

**In Section 2.5: My suggestion is to add a figure to illustrate the domain coverage and provide statistical or graphical evaluations of the modeled meteorological fields against observations. In addition, though the simulated O3 concentrations have good statistical agreements with those observed as shown in Fig S3, there were still some biases in Chengdu, Nanchong, and Deyang. Some discussions on the discrepancies are needed.**

**Reply:** Thanks for the suggestion. We have done the following revisions,

(1) We have added a figure to illustrate the domain coverage in the supplementary file. Please see Fig S2.

[Figure]

**Figs S2 Domain coverage of the WRF-CMAQ model. The outer box shows the domain of d01 and the inner box shows the domain of d02.**

(2) Evaluations of meteorological simulations including temperature, relative humidity and winds were summarized in Table S4. Generally, these meteorological parameters matched well with the observations, with high correlation coefficient and low bias. It was notable that the wind speed was a little overestimated, which

was a common issue for weather numerical simulations, especially for a grid resolution of 12×12km.

Table S4 Statistical validation of WRF simulated meteorological parameters.

|  | MB | RMSE | R |
|---|---|---|---|
| Temperature (°C) | -0.6 | 2.1 | 0.91 |
| Relative humidity (%) | -4.5 | 8.6 | 0.86 |
| Wind speed (m/s) | 1.6 | 1.8 | 0.45 |

(3) It is important to emphasize that we have included the results of the CMAQ model spin-up in our presentation. The initial five days were dedicated to model pre-warming, and as a result, there is a significant disparity between the simulated ozone and observations during this initial period. Please see our revisions, "After a spin-up of 5 days, the WRF-CMAQ model was performed to simulate $O_3$ concentrations in the SCB."

Based on the validation, it could be found out that we captured the changing trends of basin ozone and successfully replicated the simulation of summer ozone pollution as evidenced by the high value of IOA (0.78~0.85). We admit that there was difference between the simulations and observations in some details, this was mainly attributed to the highly complex topography of the Sichuan Basin, characterized by high elevation differences and deep basin topography, posing significant challenges to our numerical simulations. One method to improve

simulation results is to increase the model's resolution. Constrained by the spatial resolution of the MEIC inventory (0.25° × 0.25°), we adopted a grid resolution of 12 km × 12 km in this study. Therefore, we also call for the development of higher-resolution emission inventories in the future to enhance the performance of air quality numerical simulations.

**In Lines 251-256: The authors could further introduce how the DDM method was more refined than the BFM method.**

**Reply:** Thanks for the question. We have incorporated the following introduction in the manuscript, "In this study, we introduced the CMAQ-DDM (Decoupled Direct Method) module to investigate the non-linear relationship between $O_3$ and its precursors. Unlike the traditional brute force method (BFM) that involves cutting or eliminating emissions from source regions (or sectors), which is not only computationally intensive but also prone to uncertainties (due to the intricate non-linear nature of $O_3$ chemistry), the DDM method offers a more refined alternative. It enables accurate and computationally efficient calculations of the sensitivity coefficients required for evaluating the impact of parameter variations on output chemical concentrations (Napelenok et al., 2008). Furthermore, the DDM method has been reported to exhibit more accurate calculations when addressing uncertainties arising from the

nonlinear relationship between secondary pollutants and their emissions, in comparison to the BFM (Itahashi et al., 2015)."

**In Fig 3: Please introduce each factor in the figure caption.**

**Reply:** Thanks for the suggestion. We have added the factor in the figure caption, "Temp, SSR, BLH, U10, V10, W, $ACCO_3$ and $ACCNO_2$ stands for temperature, surface solar radiation, boundary layer height, 10 m u-component of wind, 10 m u-component of wind, vertical wind, previous night accumulative $O_3$ and previous night accumulative $NO_2$, respectively"

**In Lines 452-454: The contributions of heatwave events to isoprene emissions were quantified in two cities. Is it based on MEGAN model simulation? Please elaborate on the calculation process.**

**Reply:** Yes, we adopted MEGAN model to calculate isoprene emission. As shown in Section 2.5, we have presented the introduction of the model, "Besides, natural emissions were calculated using MEGAN model (version 2.1) driven by the WRF simulated meteorology. The static input vegetation-related data of MEGAN were updated by using the 2020-based the plant function type (PFT) and leaf area index (LAI) retrieved from the MODIS (Moderate-Resolution Imaging Spectroradiometer) products". Here, we used two sets of meteorological fields to drive the MEGAN model. One set corresponds to the meteorological fields simulated by WRF

for the summer of 2021, while the other set corresponds to the meteorological fields simulated by WRF for the summer of 2022. Among these, the BVOC emissions obtained by driving the MEGAN model with the meteorological fields from the summer of 2021 are considered as ISOP emission by base meteorology (BaseMETE). The difference in BVOC emissions obtained by driving the MEGAN model with the meteorological fields from the summer of 2022, compared to Base METE, is considered as ISOP emission induced by heatwave (ByHeatwave). We have added the illustration in the manuscript.

---

## Author Comment (AC2)

**Response to Reviewer 2**

**General Comments:**

The unprecedented heatwave of August 2022 brought about national wide $O_3$ pollution in China. This study targeted the heatwave month of August, 2022 in the air pollution basin SCB in Southwest China presenting the detailed investigation on the spatial disparity of $O_3$ pollution between two major urban areas Chongqing and Chengdu with pollution levels of $O_3$ in the western SCB (Chengdu) but relatively lower concentrations in the eastern basin (Chongqing). Meteorological and precursor factors were assessed with observation, modeling study and machine learning methods, spotlighting high temperatures, intensive solar radiation, and overnight accumulative pollutants as key contributors to $O_3$ concentration, revealing the important role of meteorology-induced natural emissions and meteorological changes. It is suggested to be accepted this paper for ACP publication in the ACP after the major revisions:

**General response to the reviewer 1:** We are grateful to the reviewer for reviewing and helping to improve our manuscript. The suggestions provided contribute to making our article more logically rigorous and enhancing its overall quality. We have followed the instructions and improved our manuscript accordingly. Hopefully, this enhanced manuscript could meet with approval. Below is our point-to-point response.

**Major comments:**
As mentioned in the discussions, if the cross-regional transport predominantly influenced Chongqing (contributing ~80%), the local photochemical $O_3$ production contributed merely about 20 % to O3 variations during August 2022 in Chongqing, where the prevailing southerly drived the transport of poor-O3 air masses from the Yungui Plateau,because Chongqing is immediately adjacent to the Yungui Plateau in the south. Please give the detailed discussions and highlighted the results on the regional transport importance in the spatial disparity of O3 pollution between Chongqing and Chengdu in the following respects of changing emission and meteorology:
a) the lower $O_3$ in Chongqing was dominated by the transport of poor-$O_3$ air masses from the Yungui Plateau with the less contribution of photochemical O3 production from local and regional transported precursors.
b) the cross-regional transport of $O_3$-precursors from the Yungui Plateau with low anthropogenic emissions and high natural emissions. It is possible to identify the local and non-local O3-precursors with their relative contribution

c) the spatial disparity of $O_3$ pollution between Chongqing and Chengdu was decided by the changes in regional transport of $O_3$ and its precursors respectively with high and low contribution of non-local $O_3$-precursors to the two urban areas.

d) Please compare the 33-yr (1990-2022) averages of air temperature, relative humidity, wind speed and direction over August between Chengdu and Chengdu with the anomalies in August,2022. Please clarify that the anomalies of high air temperature and low relative humidity are connected with the strong-southerly-driven cross-regional transport of cool and dry air masses from the Yungui Plateau, which could suppress the photochemical $O_3$ production in Chongqing.

**Reply:** We feel grateful to the reviewer for providing these valuable suggestions. We have improved the manuscript based on these suggestions.

Firstly, to illustrate the different impact of regional transport on both cities. We have provided evidence from the following four aspects.

(1) By analyzing the surface diurnal meteorological parameters including winds, BLH and VC. Generally, we found lighter winds, lower BLH and smaller VC in Chengdu compared to Chongqing.

(2) By analyzing ratio of m, p-xylene and ethylbenzene. Given that m, p-xylene is more reactive than ethylbenzene, their ratios typically decrease due to photochemical reactions that take place during the transport of air masses. As shown in Fig S6, the ratio was much lower in Chongqing (1.04 ppbv ppbv-1), indicating the presence of "aged" air masses being monitored. Conversely, a higher ratio (3.11 ppbv ppbv-1) in Chengdu indicated the prevalence of "fresh" air masses likely originating from local emissions. The discovery reaffirmed that Chongqing exhibited superior ventilation conditions compared to Chengdu.

(3) By conducting LPDM simulation. We found significant differences in the dominant air masses influencing Chengdu and Chongqing. Chengdu's air masses were predominantly influenced locally, while Chongqing's were predominantly influenced by cross-regional transport (from the Yungui Plateau).

(4) By identifying the contribution of local and non-local precursors to $O_3$ concentrations in both cities (CMAQ-ISAM simulation).

Secondly, in order to highlight the importance of the cross-region transport of air

masses from the Yungui Plateau, we improved the discussions and modified Fig 8.

[Figure]

Fig 8 (a) Distribution of monthly averaged 90th percentile of MDA8 O₃ (MDA8-90) concentrations and monthly averaged winds at 500hPa; (b) 72h retroplume (footprint residence time) showing transport pathways of air masses arriving at Chengdu, and distribution of anthropogenic NOₓ emissions; (c) 72h retroplume (footprint residence time) showing transport pathways of air masses arriving at Chongqing, and distribution of biogenic isoprene (ISOP) emissions in August 2022

The distribution of O₃ concentration in China's southwestern region, as shown in Fig 8a, revealed that high O₃ concentrations were mainly concentrated in the SCB region. In contrast, the O₃ concentration in the adjacent Yunnan-Guizhou Plateau (southeast) was very low, indicating a poor-O₃ region. According to the synoptic flows, it could be seen the prevailing wind was southeastward, and the wind speed gradually decreased from east to west, implying that Chengdu was more stationary than Chongqing. Our LPDM-simulated 72h backward retroplumes (Fig 8b) showed

that, Chengdu was primarily influenced by local air masses encompassing areas such as Chengdu city and the eastern parts of the SCB. Relatively fewer air masses originated from cross-province transport in the southeast direction. The distribution of NOx emissions showed that Chengdu was significantly influenced by the locally anthropogenic emissions. Differently, Chongqing showed a situation to be more susceptible to cross-regional transport influences (Fig 8c). The dominant air masses in Chongqing not only originated locally but also experienced cross-province transport from the southeast, influenced by the regions such as Yungui Plateau, a poor-$O_3$ region with relatively low anthropogenic emissions but high BVOC emissions. Considering the strong reactivity and limited lifetime of BVOCs, their role on downwind air quality was limited. To support this, we adopted CMAQ-ISAM to identify the local and non-local $O_3$-precursors with their relative contribution to $O_3$ concentrations in both cities. As shown in Fig 9b and Fig 9d, Chengdu was mainly affected by local regions, contributed to 46.8%. This implied that local emissions within the SCB were a significant contributor to the excessive $O_3$ levels in Chengdu. In contrast, the influence of the local region on $O_3$ levels in Chongqing was ~20%. Instead, the contribution outside the basin almost reached 50%, indicating that Chongqing was more susceptible to the influence of cross-regional transport.

Furthermore, we also added Fig S6 in the supplementary file. By examining the difference between 2022 Aug and the climate average (1990-2021), it was found that the anomalies of high air temperature and low relative humidity were connected with the strong-southerly-driven cross-regional transport of cool and relatively clean air masses from the Yungui Plateau, which could suppress the photochemical $O_3$ production in Chongqing. In contrast, Chengdu was in a typical stationary condition with light wind, high temperature and low relative humidity, which were conducive to a local photochemical pollution. In general, the spatial disparity of $O_3$ pollution between Chongqing and Chengdu was decided by the changes in regional transport of $O_3$ and its precursors respectively with high and low contribution of non-local $O_3$-precursors to the two urban areas.

[Figure]

Fig S6 (a) Distribution of temperature and winds at 1000 hPa during 2022 Aug; (b) Distribution of temperature anomaly (2022 – climate average) and the averaged wind between 1990 and 2021. (c) same as (a) but for relative humidity (RH). (d) same as (b) but for RH.

**Specific comments:**
**1) Observational data showed that Chengdu experienced a consecutive 17-day period of O3 exceedance,Did the extreme heatwave last the consecutive 17-day period in Chengdu? How long was the extreme heatwave in Chongqing? As the study targeted the month of August, 2022 (not a heatwave), please modify the title of manuscript for the study.**

**Reply:** Thanks for the questions. By using the standard that the daily maximum temperature is above 32°C (WMO standard). There are 29 heat days in Chongqing and 24 heat days in Chengdu (Fig R1). As kindly suggested, we changed the title to "Spatial disparities of ozone pollution in the Sichuan Basin spurred by extreme hot weather".

[Figure]

Fig R1 Time series of temperature in Chengdu and Chongqing during Aug 2022

**2) The first paragraph in sect. 1 Introduction is too long with the disorder presentation. Please restructure this paragraph in 2-3 paragraph.**

**Reply:** Thanks for the suggestion. We now divided this paragraph into 2 paragraph .

We believe this revised arrangement is much better.

In the revised introduction, paragraph one mainly introduces the complexity and difficulty in controlling $O_3$. Paragraph two mainly introduces the non-linear relationship between $O_3$ and its precursors. Paragraph three mainly talks about the role of meteorology in $O_3$ formation. Paragraph 4 introduces the region of SCB. Paragraph 5 introduces the study target of this research.

**3) Please mark the location of Tibetan Plateau and Yungui Plateau and indicate the meaning of color contours in Fig. 1a. Please change Fig. 1b with the 33-yr averages of air temperature over August between Chengdu and Chengdu with the anomalies in August over 1990-2022 based on the site observations.**

**Reply:** OK. We have modified Fig1 based on the comment. Please see our revised Fig 1.

[Figure]

Fig 1 (a) Geographical distribution of Sichuan Basin with scattered averaged monthly MDA8

O₃ concentrations (data obtained from Ministry of Ecology and Environment of China). The contoured shows the 3D terrain height in SCB. The black lines highlight the administrative border of Chengdu and Chongqing, respectively. The blue stars indicate the supersite in Chengdu and Chongqing, respectively. (b) Historical monthly averaged daily-maximum air temperature (August) comparison between 2022 and 1990-2021 of the SCB.

**4) Please carefully check the caption of Fig. 2 against the Figure, give a correct caption (vc, wind, averages over August 2022).**

**Reply:** Thanks. We have revised the captions as suggested.

**5) Fig. 1g presented the significant differences in RH. Please add the discussions on role of RH in the spatial disparity of O3 pollution between Chongqing and Chengdu.**

**Reply:** Usually, a condition with lower RH would be more conducive to photochemical reaction. As RH reflects the amount of water vapor in the atmosphere, which is a removal source of O₃ concentrations (e.g., through HOₓ reactions). We found that Chongqing had higher temperature, stronger solar radiation and lower relative humidity than Chengdu, but had lower O₃ concentrations. This is the phenomenon that has left us puzzled.

We revised the discussions based on the following, "*Additionally, significantly lower relative humidity was observed in Chongqing, suggesting a potential reduction in O₃ removal by water vapor, for instance, through HOₓ reactions.*"

**6) Please remove the discussions (lines 302-313) on the synoptic system. 200hPa and 500hPa are too high to discuss the stationary and unstable atmosphere for air quality change over the SCB.**

**Reply:** Thanks. We have removed this part.

**7) Please thoroughly check the English presentations in the text and all the Figs. with the clear and captions.**

**Reply:** Okay. We have carefully revised the manuscript as suggested. Revisions are highlighted in red color.